# Potentiation of Antibiotic Activity of Aztreonam against Metallo-β-Lactamase-Producing Multidrug-Resistant *Pseudomonas aeruginosa* by 3-*O*-Substituted Difluoroquercetin Derivatives

**DOI:** 10.3390/pharmaceutics16020185

**Published:** 2024-01-28

**Authors:** Seongyeon Lee, Taegum Lee, Mi Kyoung Kim, Joong Hoon Ahn, Seri Jeong, Ki-Ho Park, Youhoon Chong

**Affiliations:** 1Department of Bioscience and Biotechnology, Konkuk University, Hwayang-dong, Gwangjin-gu, Seoul 05029, Republic of Korea; tjddus98@konkuk.ac.kr (S.L.); co4121@konkuk.ac.kr (T.L.); 2Bio/Molecular Informatics Center, Konkuk University, Hwayang-dong, Gwangjin-gu, Seoul 05029, Republic of Korea; mkkim@konkuk.ac.kr (M.K.K.); jhanhn@konkuk.ac.kr (J.H.A.); 3Department of Integrative Bioscience and Biotechnology, Konkuk University, Hwayang-dong, Gwangjin-gu, Seoul 05029, Republic of Korea; 4Department of Laboratory Medicine, Hallym University College of Medicine, Chuncheon 24252, Republic of Korea; hehebox73@hallym.or.kr; 5Department of Infectious Disease, Kyung Hee University School of Medicine, Seoul 02447, Republic of Korea

**Keywords:** *Pseudomonas aeruginosa*, multidrug resistance, β-lactamase, efflux pump, multitarget inhibitor, aztreonam, combination

## Abstract

The combination of aztreonam (ATM) and ceftazidime–avibactam (CAZ-AVI; CZA) has shown therapeutic potential against serine-β-lactamase (SBL)- and metallo-β-lactamase (MBL)-producing *Enterobacterales*. However, the ability of CZA to restore the antibiotic activity of ATM is severely limited in MBL-producing multidrug-resistant (MDR) *Pseudomonas aeruginosa* strains because of the myriad of intrinsic and acquired resistance mechanisms associated with this pathogen. We reasoned that the simultaneous inhibition of multiple targets associated with multidrug resistance mechanisms may potentiate the antibiotic activity of ATM against MBL-producing *P. aeruginosa*. During a search for the multitarget inhibitors through a molecular docking study, we discovered that di-F-Q, the previously reported efflux pump inhibitor of MDR *P. aeruginosa*, binds to the active sites of the efflux pump (MexB), as well as various β-lactamases, and these sites are open to the 3-*O*-position of di-F-Q. The 3-*O*-substituted di-F-Q derivatives were thus synthesized and showed hereto unknown multitarget MDR inhibitory activity against various ATM-hydrolyzing β-lactamases (AmpC, KPC, and New Delhi metallo-β-lactamase (NDM)) and the efflux pump of *P. aeruginosa*, presumably by forming additional hydrophobic contacts with the targets. The multitarget MDR inhibitor **27** effectively potentiated the antimicrobial activity of ATM and reduced the MIC of ATM more than four-fold in 19 out of 21 MBL-producing *P. aeruginosa* clinical strains, including the NDM-producing strains which were highly resistant to various combinations of ATM with β-lactamase inhibitors and/or efflux pump inhibitors. Our findings suggest that the simultaneous inhibition of multiple MDR targets might provide new avenues for the discovery of safe and efficient MDR reversal agents which can be used in combination with ATM against MBL-producing MDR *P. aeruginosa*.

## 1. Introduction

Carbapenems belong to a class of β-lactam (BL) antibiotics, which show antimicrobial activity against many types of bacteria, including those resistant to other antibiotics [1]. Thus, carbapenems have been regarded as the last antibiotics on the shelf to fight against multidrug-resistant (MDR) bacteria [2]. However, with the increasing use of carbapenems in clinical practice, the emergence of carbapenem-resistant pathogens now poses a great threat to human health. This type of antimicrobial resistance, especially when mediated by transferable carbapenemase-encoding genes, is spreading rapidly, causing serious outbreaks and dramatically limiting treatment options [3,4].

Like other β-lactam antibiotics, carbapenems have a four-membered lactam ring that is vulnerable to hydrolysis by bacterial enzymes called carbapenemases, a class of β-lactamases [5]. Most β-lactamases are divided into four groups (Ambler classes A, B, C, and D), among which KPC (class A), OXA-48 types (class D), and class B metallo-β-lactamases (MBLs: NDM, VIM, and IMP) are the most effective in terms of the enzymatic activity and geographical spread [6,7]. Therefore, numerous efforts have been put forth to discover inhibitors against these resistance-related bacterial targets. Despite this, β-lactamase inhibitors (BLIs), such as clavulanate, sulbactam, tazobactam, avibactam, vaborbactam, and relebactam [8], have no (or at best limited) inhibitory activity against MBLs. Therefore, infections caused by MBL-expressing carbapenem-resistant Gram-negative bacteria remain the most difficult to treat [9,10,11].

In this context, aztreonam (ATM) has drawn particular interest because it is stable against enzymatic hydrolysis by MBLs, presumably owing to its characteristic monobactam scaffold. However, ATM suffers from hydrolysis by other β-lactamases, including extended-spectrum β-lactamase (ESBL) and AmpC β-lactamase, which are co-produced by most MBL-producing strains [12]. Recently, this problem has been successfully resolved by combining ATM with a BL/BLI such as ceftazidime–avibactam (CAZ-AVI), which inhibits ESBL and AmpC β-lactamases. This combination of ATM and ceftazidime–avibactam (CAZ-AVI) was shown to be highly effective in treating infections caused by NDM-producing *Enterobacteriaceae* [13,14]. Nevertheless, the ability of ceftazidime–avibactam (CAZ-AVI) to restore the bactericidal activity of ATM in MBL-producing *Enterobacterales* can hardly translate to *Pseudomonas aeruginosa* because the latter harbors numerous and varied antibiotic resistance elements [15,16,17,18,19,20,21]. Thus, colistin or polymyxin B remains the last treatment option against multidrug-resistant MBL-producing *P. aeruginosa* isolates.

In addition to the previously mentioned production of β-lactamases, *P. aeruginosa* has various resistance mechanisms, such as low outer-membrane permeability and a constitutive expression of efflux pumps, and its target mutations often work together to confer MDR to many strains [22,23,24]. Among those, an increase in drug efflux is a key mechanism that confers cross-resistance to many unrelated antibiotic classes through cooperation with other mechanisms [25,26,27,28,29,30], and ATM resistance in *P. aeruginosa* is also known to be associated with the overexpression of MexAB-OprM [31,32,33,34]. Taken together, we reasoned that concurrent inhibition of β-lactamases and efflux pumps might have a better chance of potentiating the antibiotic activity of ATM against MBL-producing *P. aeruginosa*. In this context, di-F-Q (**1**; Figure 1) drew our attention because, compared with its parent compound (quercetin, **2**; Figure 1) [35,36], di-F-Q (**1**) demonstrated significantly enhanced inhibitory activity against the efflux pumps of MDR *P. aeruginosa* [37]. Also interesting is that, albeit with low potency, quercetin (**2**) is capable of inhibiting β-lactamases [38,39,40]. Thus, it was anticipated that, based on the di-F-Q scaffold, a hereto unknown multitarget inhibitor might be designed which potentiates the antimicrobial activity of ATM against MDR MBL-producing *P. aeruginosa*.

## 2. Materials and Methods

### 2.1. Molecular Docking Study

The three-dimensional structures of MexB (PDB ID: 3W9J) [41], AmpC (PDB ID: 6DPT) [42], NDM-1 (PDB ID: 6V1M) [43], and KPC-2 (PDB ID: 7LNL) [43] were obtained from the Protein Data Bank. The compounds were modeled based on the three-dimensional structure of quercetin downloaded from PubChem (https://pubchem.ncbi.nlm.nih.gov (accessed on 3 May 2023); compound CID: 5280343), using Vega ZZ [44]. The AutoDock Vina [45,46] module in UCSF Chimera [47] was employed to dock the title compounds with MexB, AmpC, NDM-1, and KPC-2. The grid spacing was set at 1.0 Å for the target proteins, and the dimensions (in Å) of the grid map were set at X: 48.81, Y: 26.34, and Z: 48.96 for MexB; X: 30.79, Y: 14.52, and Z: 30.72 for AmpC; X: 32.91, Y: 15.25, and Z: 14.44 for NDM-1; and X: 18.97, Y: 36.13, and Z: 19.33 for KPC-2, respectively. The centers of the grid boxes were adjusted to (in Å) X: 26.9, Y: 24.87, and Z: 23.59 for MexB; X: 20.21, Y: 15.09, and Z: 20.37 for AmpC; X: 16.76, Y: 12.66, and Z: 3.87 for NDM-1; and X: 22.25, Y: 15.37, and Z: 8.33 for KPC-2, respectively. A Lamarckian genetic algorithm with default parameters was used, and self-docking studies were undertaken in which the ligands of the PDB structures were separated from their cognate receptors, and the resulting pairs were then subjected to a docking calculation. The root mean square deviation (RMSD) values of the best-scoring docked pose to the crystal structure pose were 0.67, 0.42, 0.74, and 0.97 A^o^ for 3W9J, 6DPT, 6V1M, and 7LNL, respectively. The binding modes of the ligands to the target proteins were analyzed with ChimeraX (version 1.7), using the interactive ViewDockX tool (https://www.cgl.ucsf.edu/chimerax/docs/user/tools/viewdockx.html) [48].

### 2.2. Chemicals and Reagents

The reagents required for chemical synthesis were purchased from Sigma-Aldrich (St. Louis, MO, USA), TCI (Tokyo, Japan), or Alfa Aesar (Ward Hill, MA, USA). TLC was performed on silica gel-60 F254 purchased from Merck (Rahway, NJ, USA). Column chromatography was performed using silica gel-60 (220−440 mesh) for flash chromatography. Nuclear magnetic resonance (NMR) spectra were recorded with a JNM-ECZ500R (Tokyo, Japan) at 500 MHz for ^1^H NMR and 125 MHz for ^13^C NMR, using tetramethylsilane as an internal standard. The chemical shifts were reported as s (singlet), d (doublet), t (triplet), q (quartet), qu (quintet), sept (septet), m (multiplet), or br s (broad singlet). The coupling constants are reported in hertz (Hz). Chemical shifts were reported as parts per million (δ) relative to the solvent peak. Fast atom bombardment mass spectra (FAB-MS) were obtained at the Korea Basic Science Institute (Daegu, Republic of Korea) and reported in the form of *m*/*z* (intensity relative to the base peak = 100).

Dimethyl sulfoxide (DMSO), crystal violet, phosphate-buffered saline (PBS), glucose, polymyxin B, ethidium bromide (EtBr), and carbonyl cyanide *m*-chlorophenyl hydrazine (CCCP) were purchased from Sigma-Aldrich. Cation-adjusted Mueller–Hinton broth (CAMHB) and Luria-Bertani (LB) media were obtained from BD Biosciences (New Jersey, NJ, USA). Blood agar plates (BAPs), *N*-phenyl-1-naphthylamine (NPN), and 4-(2-hydroxyethyl)-1-piperazineethanesulfonic acid (HEPES) were purchased from Hangang (Gunpo, Gyeonggi-do, Republic of Korea), Fisher Scientific (Pittsburgh, PA, USA), and BioBasic (Markham, ON, Canada).

### 2.3. Synthesis

Di-F-Q (**1**) was prepared following a previously reported procedure [49]. The title compound, 3-*O*-alkyl-di-F-Q (**3**–**31**), was synthesized from compound **1** (Appendix A).

### 2.4. Microorganisms

Twenty-one clinical isolates of *P. aeruginosa* expressing either NDM or IMP were collected from patients admitted to a university hospital in Seoul, Republic of Korea.

### 2.5. Assessment of Antibacterial Activity

The antibacterial activity (minimum inhibitory concentration (MIC)) of various antibiotics (piperacillin–tazobactam (PZP), ceftazidime–avibactam (CZA), ceftazidime (CAZ), cefepime (FEP), aztreonam (ATM), imipenem (IPM), meropenem (MEM), amikacin (AMK), gentamicin (GEN), colistin (CST), and ciprofloxacin (CIP)), as well as the title compounds (3-*O*-alkyl-di-F-Q derivatives (**3**–**31**)), was determined for all isolates in three replicates, using broth microdilution with CAMHB, according to the Clinical and Laboratory Standards Institute (CLSI)’s guidelines [50]. The DMSO stock solutions of the test compounds were serially diluted to the desired concentrations in CAMHB. Then, 10 μL of the bacterial suspension (5 × 10^5^ colony forming units (CFU)/mL) was combined with 200 μL of DMSO stock solutions of the test compounds in 96-well microtiter plates (1% DMSO, final). After the incubation of the plate at 37 °C for 24 h, bacterial growth was visibly evaluated by monitoring the turbidity of the resulting suspension, and the MICs were determined as the lowest concentrations of the test compounds that inhibited the visible growth of the bacteria.

### 2.6. Inhibition of Efflux Pumps: EtBr Accumulation

The accumulation of EtBr inside the NDM-producing (PA-038*bla*_NDM_) *P. aeruginosa* clinical isolates was monitored as previously described [51]. Briefly, the *P. aeruginosa* strain was grown in 5 mL of LB medium until it reached the mid-log phase, and the bacteria were centrifuged at 3500 rpm for 15 min. The resulting pellet was washed twice with an equal volume of PBS. The concentration of the bacterial suspension was adjusted (OD_600_ = 0.3) and added to each well of a flat-bottomed 96-well black plate (182 L). After the addition of glucose (4 µL) to the cellular suspension at a final concentration of 0.4% (*w*/*v*), 4 µL of the title compound or positive control CCCP was added to the plate. Finally, EtBr (10 µL) was also added to each well at a final concentration of 1 µg/mL, and the plate was incubated at 37 °C for 60 min, during which time fluorescence emission from EtBr was recorded (λ_ex_ 480 nm/λ_em_ 630 nm) with a 2-min interval, using a Cytation 5 imaging multi-mode reader (BioTek Instruments, Inc., Winooski, VT, USA).

### 2.7. Inhibition of β-Lactamase

Recombinant AmpC from *Escherichia coli* was produced according to a previously reported protocol [42]. KPC-2 and NDM-1 cells were purchased from LifeSpan Biosciences, Inc. (Lynnwood, WA, USA) (LS-G26598) and RayBiotech, Inc. (Peachtree Corners, GA, USA) (230-00554), respectively. Genes encoding OXA-48 (residues 23–265) and VIM-2 (residues 27–266) were custom-synthesized (Bioneer Inc., Seoul, Republic of Korea) and expressed as previously described [43,52]. The β-lactamase inhibitory activity of the title compounds was then determined according to the previously reported methods [53]. Briefly, equal volumes (100 μL) of the title compounds (70–0.1 μM in PBS) and diluted enzyme stock were mixed (final concentration 2% of DMSO in 200 μL) in 96-well plates, which were then incubated at 37 °C for 20 min. The incubated mixture was added to each well containing substrates (nitrocefin for AmpC, KPC-2, OXA-48, and VIM-2; and imipenem for NDM-1) of final concentrations of 70 μM for nitrocefin and 100 μM for imipenem. Substrate hydrolysis was monitored at 486 nm (nitrocefin) or 300 nm (imipenem) for 15 min, using a Cytation 5 imaging multimode reader (BioTek Instruments Inc., Winooski, VT, USA). At the plateau level of hydrolysis, the corresponding rates of hydrolysis were compared to those obtained without inhibitors (100%) to determine the percentage of enzyme inhibition at each concentration of the title compounds. The IC_50_ values of the compounds were derived from sigmoidal dose–response curves, using GraphPad Prism 5 (GraphPad Software, Boston, MA, USA).

### 2.8. Antibiotics-Potentiation Activity

The antibacterial activity (MIC) of aztreonam (ATM) in combination with avibactam (AVI), ceftazidime–avibactam (CAZ-AVI), CCCP, PAβN, di-F-Q (**1**), or multitarget inhibitor **27** against 21 MBL-producing *P. aeruginosa* clinical isolates was evaluated against IMP- or NDM-producing *P. aeruginosa* clinical strains by the broth microdilution checkerboard synergy assay. The bacterial inocula (5 × 10^5^ CFU/mL) in 100 μL of CAMHB in a 96-well microplate were treated with serially increased (two-fold) concentrations (0.06–512 μg/mL) of the test compounds, and ATM in horizontal and vertical directions, respectively, to make a total volume of 200 μL of broth. After incubation at 37 °C for 18–24 h, the MICs of ATM upon combination with the test compounds were visually inspected. The combination of ATM with the test compounds was analyzed by calculating the fractional inhibitory concentration index (FICI) as follows:FICI = MIC_ATM_ (in combination with the title compounds)/MIC_ATM_ (ATM alone) + MIC_test compounds_
(in combination with ATM)/MIC (test compound alone)

As the test compounds did not show any antimicrobial activity up to 512 mg/L, MIC_test compounds_ was set to 512 mg/L. Based on the FICI values, the combinations were interpreted as synergistic (FICI ≤ 0.5), additive (0.5 < FICI ≤ 1.0), indifferent (1.0 < FICI ≤ 4.0), or antagonistic (FICI > 4.0).

### 2.9. Time–Kill Assay

Bacterial suspensions of NDM-producing *P. aeruginosa* (PA-017*_bla_*_NDM_ and PA-025*_bla_*_NDM_; 10^6^ CFU/mL) were prepared from exponentially growing cells. This bacterial suspension was placed into conical tubes and incubated in the presence of ATM (1/4 MIC and 1/8 MIC), compound **27** (32 mg/L), or a combination of ATM and **27** (1/4 MIC + 32 mg/L, 1/8 MIC + 32 mg/L) in a final volume of 2 mL at 37 °C. Bacteria were counted by taking 50 μL aliquots at regular time intervals (t = 0, 1, 2, 4, 8, and 24 h), preparing serial dilutions in 0.85% NaCl solution, and plating on BAP. The data are presented as the mean of at least three independent experiments.

### 2.10. Statistical Analysis

The data are presented as the mean of at least three independent experiments. A statistical analysis was performed using Student’s *t*-test. Statistical significance was set at *p* < 0.01.

## 3. Results and Discussion

### 3.1. Clinical MBL-Producing P. aeruginosa Strains Showed High-Level Antibiotics Resistance

The growing prevalence and global spread of the multidrug resistance profiles of MBL-producing *P. aeruginosa* [54] warn of the occurrence of epidemic outbreaks. Recent antimicrobial resistance surveys in South Korea have revealed an increase in the carbapenem resistance rate of *P. aeruginosa* isolates [55]. Given the clinical importance of MBL-producing *P. aeruginosa*, we used 21 clinical isolates of *P. aeruginosa* expressing either NDM or IMP collected from patients admitted to a university hospital in Seoul, Korea (Table 1). The minimum inhibitory concentrations (MICs) of 11 antimicrobial agents (piperacillin/tazobactam, ceftazidime/avibactam, ceftazidime, cefepime, aztreonam, imipenem, meropenem, amikacin, gentamicin, colistin, and ciprofloxacin) were determined based on the CLSI guidelines. The strains exhibited complex MDR profiles, extensive drug resistance (XDR) [56], and difficult-to-treat resistance (DTR) [57] bacteria. The currently investigated clinical MBL-producing *P. aeruginosa* strains were sensitive only to colistin (CST) and showed a high level of resistance to other antibiotics, including ATM (Table 1). ATM MICs against these strains were 8–64 times higher than the CLSI antibiotic susceptibility breakpoint (1 mg/L) [50].

### 3.2. Efflux Pump and β-Lactamase Inhibitors Potentiated ATM against Highly Resistant MBL-Producing P. aeruginosa

A combination of ATM with antibiotic resistance-modulating agents (ARMAs) was attempted; however, avibactam (AVI) and ceftazidime–avibactam (CAZ-AVI) were not effective in decreasing the MIC of ATM in four representative *P. aeruginosa* strains producing NDM (PA-025 and PA-038), IMP (PA-002), and VIM (PA-003) (Table 2). The antibiotic-potentiating activities of AVI and CZA were only observed in some strains at very high concentrations (128 mg/L). It is worth noting that the efflux pump inhibitors, such as carbonyl cyanide 3-chlorophenylhydrazone (CCCP) and phenylalanine–arginine β-naphthylamide (PAβN), showed similar or better antibiotic-potentiating activity compared with AVI and CZA (Table 2). By the same token, although the previously reported *P. aeruginosa* efflux pump inhibitor di-F-Q (**1**) [37] was as potent as CCCP or PAβN, **1** showed ATM-potentiating activity at significantly lower concentrations (8–16 mg/L; Table 2). Interestingly, the triple combination ATM-CCCP-AVI showed the most potent antimicrobial activity against the two highly ATM-resistant MBL-producing *P. aeruginosa* strains, albeit at very high concentrations (256 mg/L; Table 2).

This result indicates that the efflux pump inhibitors may work in concert with the inhibitors of the β-lactamase to recover the antimicrobial activity of ATM against highly resistant MBL-producing *P. aeruginosa*. Thus, it would be favorable to identify agents capable of exerting inhibitory activity against efflux pumps and β-lactamases combined into one molecule, and the identification of a multitarget inhibitor is anticipated to provide new avenues for the discovery of ARMAs, potentiating the activity of ATM against MBL-producing *P. aeruginosa*.

### 3.3. 3-O-Substituted di-F-Q (***1***) Showed Multitarget Inhibitory Activity against Efflux Pumps and β-Lactamases

In this context, quercetin drew our attention because of its potential to inhibit both efflux pumps [35,36,39,58] and β-lactamases, including ESBL [59], OXA-48 [38], and MBLs [40]. In addition, we previously demonstrated that the structural modification of the quercetin scaffold could enhance its bioactivity, and di-F-Q (**1**) showed significant efflux pump inhibitory activity against MDR *P. aeruginosa* [37]. In this study, to further increase the multitarget inhibitory activity, we initiated the derivatization of **1** based on the analysis of its binding modes to the efflux pump and β-lactamases. The three-dimensional structures of the MexB component of the tripartite efflux pump in *P. aeruginosa* (PDB ID: 3W9J) [41], as well as β-lactamases capable of hydrolyzing ATM, such as AmpC (PDB ID: 6DPT) [42] and KPC-2 (PDB ID: 7LNL) [43], were used to examine the binding modes of **1**. After the generation of the most stable conformation using Vega ZZ [44], the molecular docking of **1** into the inhibitor binding sites of the target structures was attempted using AutoDock Vina [45,46] incorporated into UCSF Chimera [47]. As shown in Figure 2A, the binding mode of **1** to the efflux pump (MexB; PDB ID 3W9J) showed that its 3,4-difluorophenyl ring is located in the characteristic hydrophobic trap [60] of MexB, composed of Phe178, Phe610, and Phe628, thus explaining the inhibitory activity of **1** against *P. aeruginosa* efflux pumps [37]. In addition, a highly hydrophobic environment (Phe136, Tyr327, Phe615, and Phe617; colored circles) was observed around the 3-OH functionality (arrow in Figure 2A) of the di-F-Q scaffold. In AmpC hydrolase (PDB ID 6DPT) and KPC-2 (PDB ID 7LNL), the 3-OH groups of **1** were positioned in the vicinity of hydrophobic residues, such as Tyr221/Val211 and Thr237/Thr216, respectively (Figure 2B,C). Taken together, the introduction of a hydrophobic substituent at the 3-*O* position of **1** was anticipated to increase the binding affinity of the resulting derivatives to efflux pump (MexB) and ATM-hydrolyzing enzymes (AmpC β-lactamase and KPC-2), which collectively would serve to potentiate the antibiotic activity of ATM against the MBL-producing MDR *P. aeruginosa*.

### 3.4. Syntheses of the Title Compounds, 3-O-Alkyl-di-F-Q Derivatives (***3***–***31***)

The preparation of a series of 3-*O*-alkyl-di-F-Q analogs (**3**–**31**) was accomplished starting from di-F-Q (**1**) (Figure 1; Appendix A).

### 3.5. Compounds ***23***, ***26***, and ***27*** Showed Concentration-Dependent Efflux Pump Inhibitory Activity in MBL-Producing P. aeruginosa

The title compounds, 3-*O*-alkyl-di-F-Q analogues (**3**–**31**), that were obtained were evaluated for their ability to inhibit the efflux pump activity of MBL-producing *P. aeruginosa* clinical isolates. EtBr (ethidium bromide) is an efflux-prone fluorescent probe that emits fluorescence once it accumulates inside bacterial cells, owing to its binding to periplasmic components [61]. In the *P. aeruginosa* clinical isolates studied (PA-038*bla*_NDM_), EtBr florescence became stronger upon the addition of the efflux pump inhibitor CCCP to the culture media (Figure 3A), indicating that the inhibition of efflux contributes to the attenuated accumulation of EtBr in these strains. As anticipated, **1** was as potent as CCCP in increasing EtBr fluorescence in the NDM-producing *P. aeruginosa* strain PA-038*bla*_NDM_ (Figure 3A). Interestingly, the treatment of PA-038*bla*_NDM_ with di-F-Q derivatives (**23**, **26**, and **27**) resulted in a significant increase in EtBr accumulation (Figure 3A), and EtBr fluorescence in PA-038*_bla_*_NDM_ increased in a dose-dependent manner with increasing concentrations of **27** (Figure 3B). However, as accumulation reflects a balance between influx and efflux, the inhibition of the efflux pump activity of **27** was further evaluated by monitoring EtBr fluorescence from EtBr-pretreated PA-038*bla*_NDM_ (Figs. 3C and 3D). The EtBr fluorescence observed in the absence of **27** decreased in a time-dependent manner (DMSO; Figure 3C), while the addition of **27** prevented this decrease in EtBr fluorescence (Figure 3C). The concentration-dependent inhibition of efflux pump activity by **27** (Figure 3C) was represented by an IC_50_ of 10.4 mg/L (Figure 3D). In addition to **27**, other di-F-Q derivatives, such as **23** and **26** (Figure 3A), showed efflux pump inhibitory activity, but the structure–activity relationship (SAR) was noteworthy; only the di-F-Q derivatives with an ethylene-tethered steric bulk at their 3-*O* positions (**23**, **26**, and **27**) were identified as efflux pump inhibitors. Neither a heteroatom (O or N) in the linker nor a insufficient steric bulk is allowed in the active analogs, and this fact is in good agreement with the highly hydrophobic nature of the ligand-binding site of MexB (Figure 2A).

### 3.6. Compounds ***23***, ***26***, and ***27*** Exhibited Broad-Spectrum β-Lactamase Inhibitory Activity

The title compounds were evaluated for their ability to inhibit β-lactamase activity. As ATM is known to be hydrolyzed by AmpC [12], as well as carbapenemases, such as KPC [62] and NDM [63], the influence of the title compounds on the enzymatic activity of AmpC and four different carbapenemases (KPC-2 (class A), OXA-48 (class D), VIM-2, and NDM-1 (class B)) was investigated (Table 3). These enzymes were either expressed (AmpC [42], OXA-48 [43], and VIM-2 [52]) or purchased (NDM-1 and KPC-2), and their activities were monitored using nitrocefin (AmpC, KPC-2, OXA-48, and VIM-2) or imipenem (NDM-1) as fluorescent substrates [43]. Di-F-Q (**1**), previously determined not to inhibit ESBLs [37], did not show inhibitory activity against the β-lactamases studied (Table 3). However, as anticipated by the molecular docking study, significant broad-spectrum β-lactamase inhibitory activity was observed in some 3-*O*-substitued di-F-Q derivatives, such as **23**, **26**, and **27**. Interestingly, the efflux pump inhibitors **23**, **26**, and **27** (Figure 3A) were also identified as pan-β-lactamase inhibitors, which suggests that the bulky hydrophobic groups tethered at the 3-*O* position via an ethylene linker play a key role in binding to both efflux pump and β-lactamase. Moreover, this SAR was more strictly observed in AmpC inhibitors, and compounds with shorter or longer linker (**6–7** and **28–29**) or less bulky substituents (**24** and **25**) failed to inhibit AmpC, although they inhibited broad-spectrum carbapenemase activity (Table 3).

### 3.7. Molecular Docking Study of ***27*** Showed the Binding Role of Its 3-O-Substituent to the Efflux Pump and the Broad-Spectrum β-Lactamases

Taken together, the efflux pump inhibitors identified in this study (**23**, **26**, and **27**) showed simultaneous inhibitory activity against a wide range of β-lactamases, including those capable of hydrolyzing ATM (Table 3). The unknown simultaneous inhibition mechanism and the underlying binding sites in the efflux pumps and the β-lactamases deserve to be explored in more detail, which may serve as a novel approach for the discovery of more potent MDR modulators. As the mulitarget MDR inhibitors (**23**, **26**, and **27**) share similar structural features, a steric bulk separated by an ethylene moiety from 3-*O* of di-F-Q scaffold, the possible role of the 3-*O* substituents in the binding of the di-F-Q derivatives to both efflux pumps, as well as β-lactamases, needed to be investigated. Thus, the characteristic binding modes of the multitarget inhibitor **27** to efflux pumps and β-lactamases were analyzed using molecular docking studies. In the MexB structure, **27** was correctly recognized by the phenylalanine-rich hydrophobic trap [60], with its di-F-Q core structure in close contact with Phe178, Phe610, Phe615, and Phe628 (Figure 4A). The characteristic 3-*O*-cyclohexylethyl substituent of **27** fit well into the remaining part of the trap composed of Phe136, Tyr327, Val571, Phe573, and Met630 (Figure 4A). On the other hand, as shown in Figure 2, β-lactamases, such as AmpC, KPC-2, and NDM-1, have a common ligand binding pocket that is distinct from the enzymatic active site (i.e., an exosite), and these sites are not shown to be occupied by currently available BLIs. However, **27** snuggly fit across both sites, and its 3-*O*-cyclohexylethyl substituent was located at the unexplored binding sites, whereas the di-F-Q core mimicked the key target-binding interactions of the BLIs (Figure 4B–D). Specifically, the boronic acid-mediated metal binding interaction of QPX7728, a broad-spectrum BLI against NDM-1 [43], was reproduced by the two oxygen atoms of **27** (4-CO and 5-OH; Figure 4D).

### 3.8. Antimicrobial Activity of ATM Was Significantly Increased by Combination with ***27***

As such, the 3-*O*-alkyl substituents of the multitarget inhibitors (**23**, **26**, and **27**) increased the binding affinity to both efflux pumps (MexB; Figure 4A) and β-lactamases (AmpC (Figure 4B), KPC-2 (Figure 4C), and NDM-1 (Figure 4D)) by forming additional hydrophobic contacts. Of particular interest is that, even though ATM resistance in *P. aeruginosa* is known to be associated with overexpressed MexAB-OprM [32,33,34], a combination of efflux pump inhibitors (EPIs) with ATM was not evaluated as a treatment option for infections caused by MDR MBL-producing *P. aeruginosa*. By the same token, the use of multitarget inhibitors as the ATM-potentiating agent against this formidable pathogen has yet to be evaluated. Thus, a checkerboard assay was performed, and the antibiotic-potentiating activity of **27** was evaluated in 21 clinical isolates of NDM- or IMP-expressing *P. aeruginosa* isolates in combination with ATM (Table 4). Compound **27** alone did not show antimicrobial activity at concentrations up to 512 mg/L. In addition, no cytotoxic effects were observed when the human liver (HepG2) cell line was treated with **27** (up to 64 mg/L; Appendix A Appendix A). However, upon combination with ATM, the multitarget MDR inhibitor **27** was effective in reversing antimicrobial resistance and showed more than a four-fold MIC reduction (synergistic effect) of ATM in 19 out of 21 strains (90%; Table 4). In particular, the synergistic effects of the ATM-**27** combination were observed in 15 out of 16 NDM-producing *P. aeruginosa* strains (94%; Table 4). In contrast, these strains were highly resistant to combinations of ATM with BLIs and/or EPIs (Table 4), and neither AVI nor CCCP was effective in decreasing the MIC of ATM upon combination treatment; only marginal potentiation of the antimicrobial activity of ATM by **27** was observed in a limited number of *P. aeruginosa* strains (Table 4). Collectively, these results demonstrate that the numerous and varied antibiotic resistance mechanisms associated with *P. aeruginosa* [15,16,17,18,19,20] cannot be overcome by the inhibition of a single resistance mechanism, and the simultaneous inhibition of multiple MDR targets might provide new avenues for the discovery of safe and efficient MDR reversal agents.

Since previous studies reported that the disruption of quorum sensing enhanced sensitivity to traditional antibiotics [64], we evaluated the synergistic anti-biofilm activity of the ATM/**27** combination. When MBL-producing *P. aeruginosa* static biofilms were treated with 1/2 × MIC ATM alone, no significant inhibition of biofilm formation was observed. The addition of **27** (32 mg/L) was not effective in enhancing the anti-biofilm activity of ATM, and inhibition of biofilm formation was observed in only 4 out of 21 MBL-producing *P. aeruginosa* isolates studied (OD_600_ by 24–82%; Appendix A Appendix A). Thus, the lack of consistent synergistic killing of static biofilm suggests that the ATM-potentiating activity of **27** in MBL-producing *P. aeruginosa* is not likely to be associated with its quorum-sensing inhibition.

### 3.9. Antibiotic-Potentiating Activity of ***27*** Was Confirmed by Time–Kill Assay

This synergy was also demonstrated by the time–kill curves for PA-017, *bla*_NDM_ (Figure 5A) and PA-025, *bla*_NDM_ (Figure 5B). In both strains, the combination of ATM (1/2 MIC) and **27** (64 mg/L) effectively lowered the exponentially increasing population of NDM-producing *P. aeruginosa* and reductions in CFU/mL of over 3-log_10_ were observed at 24 h relative to the starting inoculum.

## 4. Conclusions

Gram-negative bacteria with carbapenem resistance pose one of the greatest threats to human health, wherein the key mechanism conferring resistance to carbapenems is the production of β-lactamases. Unlike SBLs, MBLs are difficult to inhibit, and the development of treatment options for infections caused by MBL-producing Gram-negative bacteria are urgently required. Because of its characteristic monobactam scaffold, ATM evades hydrolytic disintegration by MBLs. Unfortunately, most MBL-producing strains co-produce ESBLs, AmpC β-lactamases, and other β-lactamases that hydrolyze ATM. Recently, studies have explored the utility of ATM in combination with BL/BLIs to exploit their unique characteristics. With this approach, a BL/BLI such as CAZ-AVI (CZA) inhibits ESBLs and AmpC β-lactamases, whereas ATM evades MBL-mediated hydrolysis and exerts its bactericidal effects. Indeed, the combination of ATM and CZA is highly synergistic against NDM-producing Enterobacteriaceae species. Based on this in vitro activity and the lack of alternative treatment options, the combination of ATM-CZA is increasingly being used clinically. However, the myriad of intrinsic and acquired resistance mechanisms among multidrug-resistant *P. aeruginosa* severely limits the ability of CZA to restore ATM activity. In particular, efflux pump-mediated ATM resistance has been reported. Thus, the combination of efflux pump inhibitors (EPIs) with ATM or ATM-CZA may be a viable option for the treatment of multidrug-resistant (MDR) MBL-producing *P. aeruginosa*. In this study, we evaluated the MDR-modulating activity of a previously reported *P. aeruginosa* efflux pump inhibitor, di-F-Q (**1**), and its derivatives in clinical strains of MBL-producing *P. aeruginosa*.

Surprisingly, di-F-Q derivatives with an alkyl chain at its 3-*O* position (**23**, **26**, and **27**) showed hereto unknown multitarget MDR inhibitory activity against various ATM-hydrolyzing β-lactamases (AmpC, KPC, and New Delhi metallo-β-lactamase (NDM)), as well as the *P. aeruginosa* efflux pump, presumably by forming additional hydrophobic contacts with the targets. Upon combination with ATM, the multitarget MDR inhibitor **27** was effective in reversing antimicrobial resistance and showed an MIC reduction of over four-fold (synergistic effect) of ATM in 19 out of 21 MBL-producing *P. aeruginosa* clinical strains. In particular, the synergistic effects of the ATM-**27** combination were observed in 15 of 16 NDM-producing *P. aeruginosa* strains that were highly resistant to various combinations of ATM, BLIs, and/or EPIs. In contrast, neither AVI nor CCCP were effective in decreasing the MIC of ATM upon combination treatment, and only a marginal potentiation of the antimicrobial activity of ATM by these compounds was observed in a limited number of *P. aeruginosa* strains. Collectively, these results demonstrate that the numerous and varied antibiotic resistance mechanisms associated with *P. aeruginosa* cannot be overcome by the inhibition of a single resistance mechanism, and the simultaneous inhibition of multiple MDR targets may provide a novel strategy to combat the ever-increasing threat of antibiotic resistance conferred by MBL-producing *P. aeruginosa*.

## Data Availability

The data are contained within the article.

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
