# Peer review of "Potentiation of Antibiotic Activity of Aztreonam against Metallo-β-Lactamase-Producing Multidrug-Resistant Pseudomonas aeruginosa by 3-O-Substituted Difluoroquercetin Derivatives"

_pharmaceutics, 2024, doi:10.3390/pharmaceutics16020185_

Round 1
Reviewer 1 Report
Comments and Suggestions for Authors
Review Report
The Authors of the manuscript contributed valuable insights into the evaluation of the multidrug resistant (MDR)-modulating activity of the Pseudomonas aeruginosa efflux pump inhibitor a 3’,4’-difluoro derivative of quercetin (di-F-Q) and its derivatives against clinical strains of metallo-β-lactamases (MBL)-producing P. aeruginosa. Nowadays, in the era of increasing antimicrobial resistance of human pathogens, new options for the treatment of infections caused by MBL-producing Gram-negatives are urgently required. The results of this study provide a novel strategy to combat the ever-increasing threat of antibiotic resistance posed by this group of bacteria. Generally, the manuscript has been meticulously edited in accordance with the instructions for authors and includes all the necessary elements and sections expected from a well-prepared scientific work. Literature is relevant and appropriately selected. All literature items have been cited in the text. Nevertheless, I have noticed some points and aspects (listed below) that require editing and corrections.
Abstract
• Line 26: For “Enterobacterales”, use italics. Introduction
• I kindly recommend that You significantly shorten the text in this section.
• Line 79, 80, 82 and in other sections line 220, 274: I suggest using the name of the active substance (ceftazidime-avibactam; CAZ-AVI) instead of the name of the commercial drug (Avycaz).
• Line 81: For “Enterobacteriaceae”, use italics.
• Line 83: For “Enterobacterales”, use italics.
Materials and Methods
• Line 135, 136: After MexB, AmpC, NDM-1 insert semicolons instead of commas.
• Line 138, 139: After MexB, AmpC, NDM-1 insert semicolons instead of commas.
• Line 142: I suggest adding a reference.
• Line 144, 145: Add information about the origin (country, state, city) of the substance or reagent, when it is first used (Sigma-Aldrich, TCI, Alfa Aesar, Merck).
• Line 157: In this case, remove the name of the country, state and city.
• Line 229, 230: I suggest writing this calculation as a separate math formula. • Line 233: Please delete second “F” in (FICIF≤0.5).
Results and Discussion
• Line 266: Please, remove the dot at the end of the table title.
• Line 331: Please, remove the dot at the end of this sentence.
• Line 333, 334: I suggest increasing the font size (applies to the entire sentence).
• Lines 340 – 361: I suggest increasing font size (applies to the entire paragraph).
• Line 352: Please, remove the dot at the end of the “Figs.”.
• Line 366: Please, remove the dot at the end of this sentence.
• Line 371: Should be: OXA-48 (class D) instead of the (class C).
• Table 3: Please explain or expand abbreviation “Cmpd”.
• Line 409: Please remove the dot at the end of the “Figs.”.
• Line 418: Please, remove the dot at the end of this sentence.
• Line 448: Please, delete second “F” in FICIF≤0.5.
• Line 460: Please, remove the dot at the end of this sentence.
Supplementary Material
• In my opinion the names (numbers) of Di-F-Q derivatives should not be written in italics (e.g. line 46: (3); line 60: (4); line 72: (5); …; and line 418: (31).
References
• Line 523: For “Enterobacterales”, use italics.
• Line 534: For “Enterobacter”, use italics.
• Line 535: For “Klebsiella pneumoniae”, use italics.
• Line 538: Please, in “Bonomo, R. A..” remove one of the dots.
• Line 539: Please, in “Antimicrob .Agents” reorder dot with space.
• Line 549: Please, remove the dot and space just before “Periasamy”.
• Line 557: For “in vitro”, use italics.
• Line 562: For “Pseudomonas”, use italics.
• Line 570: For names of the genes, use italics.
• Line 572: For “In vivo”, use italics.
• Line 598: Please, in “APMIS.” remove the dot.
• Line 631: For “Pseudomonas aeruginosa”, use italics.
• Line 663: For “Klebsiella pneumoniae”, use italics.
Dear Authors, in my opinion, You have introduced a very interesting, innovative, promising and valuable work. After minor revision, I will consider this manuscript as qualified for publication. Best Regards

Comments on the Quality of English LanguageMinor editing of English language is required.
Author Response
Response to the reviewer 1's comments is attached (Revision Letter-1.pdf).

Reviewer 2 Report
Comments and Suggestions for Authors
This study explores compounds with antibacterial activity through molecular docking research and chemical synthesis, aiming to discover new agents against drug-resistant bacteria. While the study design is reasonable, there are some suggestions:
1. **Abstract Length:**
The abstract is overly lengthy; consider a more concise version summarizing key findings and objectives.
2. **Introduction Precision:**
In the introduction, the sentence about Gram-negative bacteria should be refined to state that their acquisition of carbapenem-resistance and global dissemination pose one of the significant threats to human health.
3. **Figure 3 Presentation:**
Consider revising the presentation format of the ABC image in Figure 3 for better organization and clarity.
4. **Safety Evaluation for Compound 27:**
Conduct additional safety evaluation tests, such as hemolysis and cytotoxicity tests, for compound 27 to ensure comprehensive safety assessment.
5. **Professional English Revision:**
It is recommended to have the manuscript professionally revised by an English language expert to enhance overall language quality and clarity.
Author Response
Responses to the reviewer 2's comments are attached (Revision Letter-2.pdf).

Reviewer 3 Report
Comments and Suggestions for Authors
“Potentiation of antibiotic activity of aztreonam against metallo-β-lactamase-producing multidrug resistant Pseudomonas aeruginosa by 3-O-substituted difluoroquercetin derivatives”
Antimicrobial resistance poses a severe threat, this study is extensive and meticulous quality work. As for the experimental design and the innovative work I have few comments, these are only observations in terms of understanding the text and how to describe their results, I hope my comments serve to improve the reading of these and highlight them as they were without doubt of great relevance.
Abstract
ü I recommend that authors avoid repetition, as they written in conclusion part.
Introduction
ü Current scenarios for potential effects of the biofilm and quorum sensing in antimicrobial resistance. P. aeruginosaproduces various extracellular products, such as proteases etc., inducing tissue damage and aiding infection spread. The regulation of these products involves a complex hierarchical quorum sensing cascade, and positively regulated impacting biofilm formation, could you please provide some link between these and your studies.
Methodology
ü Well structured, only concern about Molecular docking validation, did you find redocking it check the binding site.
Results and discussion
ü If the authors decided to combine both sections, it should be elaborated, the authors should critically discuss their results, not only provide the few comparative data and self-reported data. The comparison of data obtained in the present study to published data should be added along with the clear description of what new the present study adds to the current knowledge, a more detailed exposition of the points for a deeper understanding would be desirable.
Comments on the Quality of English Language
Minor editing of English language required
Author Response
Responses to the reviewer's comments are attached (Revision Letter-3.pdf).

Round 2
Reviewer 2 Report
Comments and Suggestions for Authors
No further comment